# Sustainable Agri-Food Processes and Circular Economy Pathways in a Life Cycle Perspective: State of the Art of Applicative Research

Teodora Stillitano, Emanuele Spada, Nathalie Iofrida * , Giacomo Falcone  and Anna Irene De Luca 

Department of Agriculture, Mediterranean University of Reggio Calabria, 89122 Reggio Calabria, Italy; teodora.stillitano@unirc.it (T.S.); emanuele.spada@unirc.it (E.S.); giacomo.falcone@unirc.it (G.F.); anna.deluca@unirc.it (A.I.D.L.)
* Correspondence: nathalie.iofrida@unirc.it

**Abstract:** This study aims at providing a systematic and critical review on the state of the art of life cycle applications from the circular economy point of view. In particular, the main objective is to understand how researchers adopt life cycle approaches for the measurement of the empirical circular pathways of agri-food systems along with the overall lifespan. To perform the literature review, the Preferred Reporting Items for Systematic Reviews and Meta-Analyses (PRISMA) protocol was considered to conduct a review by qualitative synthesis. Specifically, an evaluation matrix has been set up to gather and synthesize research evidence, by classifying papers according to several integrated criteria. The literature search was carried out employing scientific databases. The findings highlight that 52 case studies out of 84 (62% of the total) use stand-alone life cycle assessment (LCA) to evaluate the benefits/impacts of circular economy (CE) strategies. In contrast, only eight studies (9.5%) deal with the life cycle costing (LCC) approach combined with other analyses while no paper deals with the social life cycle assessment (S-LCA) methodology. Global warming potential, eutrophication (for marine, freshwater, and terrestrial ecosystems), human toxicity, and ecotoxicity results are the most common LCA indicators applied. Only a few articles deal with the CE assessment through specific indicators. We argue that experts in life cycle methodologies must strive to adopt some key elements to ensure that the results obtained fit perfectly with the measurements of circularity and that these can even be largely based on a common basis.

**Keywords:** systematic literature review; agricultural sustainability assessment; circular economy; lice cycle methodologies; agri-food sustainability

## 1. Introduction

### 1.1. Theoretical Background of Circular Economy

Circular economy (CE), also intended with the synonymous "circularity", is an expression that, although it is now widely used and known, remains shrouded in an aura of mystery, especially if the intent in its use is to grasp the most practical advantages in its application. For this reason, and others, discussion themes about CE are strongly explored at various levels and from different perspectives by researchers, academics, politicians, practitioners, and entrepreneurs. The CE concept, which can be dated from the original and renowned idea of "closing circle" [1], has been brought back to the forefront in 2010 due to the popular activity of the Ellen MacArthur Foundation [2], which reprised, among others, the most recent cradle-to-cradle approach [3]; since then, insights on CE never stopped moving forward. However, as Borrello et al. [4] argued, the originality of new contributions is not always clear, which risks making the concept even more disorienting, although it is shareable to consider CE nowadays as a necessary concept precisely because it is still "essentially contested" [5]. One of the key issues that make the CE discourse particularly complex is the understanding of the link between circularity and sustainability [4,6,7]

and, although it is quite shared the vision of CE as an effective way to achieve some of the sustainability goals, often the boundaries of these two overblown terms are not so see-through and this risks to blur their meaning as in a real tangle of buzzwords [8,9]. In concise and effective terms, the CE model, opposed to the linear economic model, would reduce and/or avoid resource depletion, wastes, and other environmental impacts all over the life cycle of services and products, by preserving and/or improving socioeconomic conditions. Just to provide one among the countless existing definitions, the one formulated by Kirchherr et al. [9] (p. 224) probably represents the most comprehensive: "CE describes an economic system that is based on business models which replace the 'end-of-life' concept with reducing, alternatively reusing, recycling and recovering materials in production/distribution and consumption processes, thus operating at the micro level (products, companies, consumers), meso level (eco-industrial parks) and macro level (city, region, nation and beyond), with the aim to accomplish sustainable development, which implies creating environmental quality, economic prosperity and social equity, to the benefit of current and future generations". Overlooking the theoretical and conceptual debate, most scholars focus on the need to explore what methodologies, metrics, and indicators are most suitable for evaluating the CE in the light of sustainability principles and/or dimensions, considering that a CE scenario is not necessarily more sustainable than a linear one regardless [10,11]. Furthermore, it is crucial to appropriately measure the environmental, economic, and social impacts of CE strategies and to investigate the implications of CE at different system levels, for several subjects involved by including potential rebound effects that are reflected on other production/consumption systems or other levels [12,13]. Despite the existence of varied approaches, methods, and tools to evaluate CE, scholars agree that ambiguity remains about defining the meaning of CE performances, its levels, its spatial and temporal scales, and its dimensions to be measured. Lastly, one could assume that it is probably forced to refer to a common and valid recipe for all CE contexts, the ultimate goal is probably to figure out how to join forces and use methods and tools, appropriate for specific settings, in a holistic, integrated, and complementary way. As Walzberg et al. [14] argue, a multidisciplinary cross-combination of methods could be an effective solution for CE to hybridize different metrics, extend the scope of the analysis, conduct predictive estimates of consequences, and, by using multicriteria techniques, choose the best alternatives or make trade-off choices under conditions of uncertainty and disagreement.

### 1.2. Circular Economy in a Life Cycle Perspective for the Agri-Food Sector

CE is about the rethinking of the current models of production and consumption and agri-food systems, which are responsible for the pressure on the living environment and for assuring the survival of many farms in rural areas, must necessarily move toward transition pathways. The importance of introducing CE strategies in the agri-food sector is primarily based on the circumstance—regrettably well recognized—that among the main contributors to pollution worldwide are livestock and crops, in addition to the waste production caused by downstream links in the food sector. According to European Environment Agency (EEA) [15], the food system could be considered the most defenseless of all, due to the exponential growth of total demand for food, feed, and fiber, against a relentless decline of arable land. The potential interdependencies—direct or indirect—in this context are innumerable, for example, in terms of resources competition for food or bioenergy production that requires land, energy, and water resources, or in terms of food losses and food waste that entail a value lost in supply chains, which in turn is linked to avoidable environmental impacts and financial losses [16]. Hence, the need to improve the resource efficiency of agri-food system activities should also be addressed through technical innovations to ensure more sustainable use of renewable resources, the reduction of environmental damages, and the depletion of non-renewable resources. CE application is widespread in agri-food sectors [17] because it tries to solve embedded and systemic problems, such as, for example, the conversion of waste into bio-products, new materials, or products to extending the end-of-life by generating new economic returns

or costs reduction and anyway by reducing environmental damage or optimizing the use or resources returning to the original process [18]. Therefore, it is not wrong to say that CE is potentially able to contribute to the sustainability of agri-food systems; it is about understanding how and, also means understanding how CE can help to improve specific social, economic, and environmental aspects of sustainability. Undertaking these different dimensions is methodologically challenging and calls into question the epistemological foundations of sustainability science and CE. One of the greatest concerns is the combination of different assessment methods and merging their results in a suitable and believable way. Furthermore, evaluating CE strategies should require a systemic and synergistic approach by considering the agri-food supply chain as a whole, especially to not incur the risk of making effective only one stage nor only single portions while neglecting the others [19,20]. This would mean, for example, to include the analysis of pre-production and consumption stages, in addition to co-products markets, and other secondary supply chain articulations. To satisfy these purposes, sustainability evaluation methods and, among them, the life cycle (LC) approaches, are particularly appreciated as a robust, science-based, and useful tool to measure and validate CE assumptions, help the feasibility of its implementation by receiving feedback for improvements and, finally, to communicate innovation strategies [21]. Life cycle assessment (LCA), in particular, is widely regarded to be the tool "par excellence" when it comes to evaluating the environmental impacts of circular-based products or systems [13,22]. The flexibility of LCA is also well appreciated principally because it allows incorporating it in several other metrics by demonstrating feasibility and usefulness for CE purposes [23]. However, all LC methodologies, LCA (or environmental life cycle assessment E-LCA), life cycle costing (LCC), social life cycle assessment (S-LCA), while are obtaining a growing consensus in the appraisal of different agricultural and food systems by measuring environmental, economic, and social impacts, separately or jointly, lastly required also to be systemic, multidisciplinary, and multicriterial. In these terms, the use of an LC framework, able to capture potentially all sustainability dimensions, can be adapted to evaluate CE strategies operationally and comprehensively, by shifting from the typically "cradle-to-grave" to "cradle-to-cradle" circular vision. Furthermore, to avoid partial and compartmentalized analyses in CE context, Life cycle sustainability assessment (LCSA) [24,25] is also recommended by Niero and Hauschild [19] because it could suggest elements of integration among sustainability dimensions, life cycle stages, and interdependent subjects of the supply chains by preventing or avoiding burden shifting.

### 1.3. Goal and Scope of the Review

To the best of our knowledge, within the extensive scientific literature that investigated definitions of CE, discourse typologies, applications, and measurements, no recent review has explored the use of life cycle (LC) approaches to measure the impacts deriving from the implementation of CE strategies in the context of agricultural and food productions. Only two reviews addressed a somehow related theme [17,26], but the first is mainly focused on general trends in CE research related to the agri-food sector, while the second is particularly centered on bioenergy agricultural practices. Both the studies mention LC as a tool especially desired and appropriate for CE purposes without addressing the limitations and advantages of using the tool to explain the impacts of circularity pathways.

Therefore, the originality in the scope and approach of this review is to understand how and how much the LC-based analysis is useful to evaluate if CE strategies are more sustainable than linear/traditional economic models in agri-food production systems. To address this issue, the following research questions will be answered: (1) how do researchers apply LC methods to evaluate environmental, economic, and social consequences of agri-food circular processes?; (2) how are LC methods combined with other approaches in CE measuring?; and (3) have impact results been used to increase understanding of the sustainability implications of CE strategies? This study aims to contribute to the research on CE implementation by providing an understanding of the role in life cycle approaches to measure the effectiveness of CE strategies for improving agri-food production processes

sustainability. A visual diagram is provided in Figure 1 to summarize how the CE vision, i.e., the well-known butterfly diagram by the Ellen MacArthur Foundation [2], can be brought back to a life cycle perspective through the necessary flow of data and information to measure environmental, economic, and social impacts. The paper framework is as follows: the next section describes the research methodology used in this study to conduct the systematic and critical literature review. Section 3 presents the results in terms of the main criteria used in the analysis. Sections 4 and 5 argue the discussions concerning the above-mentioned research questions and draw the research conclusions and future research proposals.

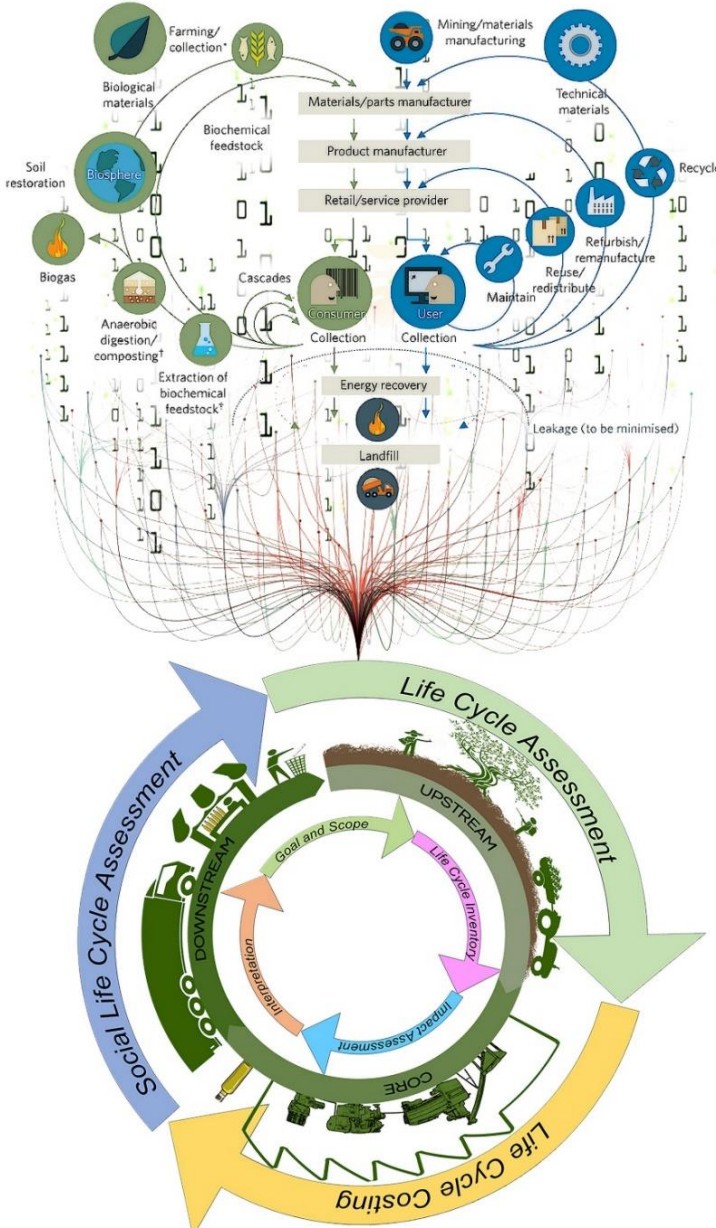

**Figure 1.** Data flow for the assessment of circular models in a life cycle perspective (our elaboration from the Ellen MacArthur Foundation) [2].

## 2. Materials and Methods

### 2.1. Literature Review

In order to provide a comprehensive vision on how much and how well life cycle methodologies are suitable to comply with CE requirements in the agri-food sector, a

systematic and critical review of the existing scientific literature was carried out. Based on the study conducted by Grant and Booth [27], a critical review goes beyond a mere description of the literature, but it should extensively evaluate its quality seeking to identify the most significant items, analyzing significant components and synthesizing the main concepts. Embracing the same main characteristics, a systematic review differs from the previous one because it seeks to systematically search for, appraise, and synthesize research evidence. Therefrom, this study, combining the strengths of these two review typologies, carry out an extensive review employing the Preferred Reporting Items for Systematic Reviews and Meta-Analyses (PRISMA) statement [28]. PRISMA was used as a formal systematic review guideline for data collection, providing a standard peer accepted methodology, to contribute to the quality assurance of the revision process and its replicability. A review protocol was developed (Figure 2), describing the search strategy, article selection criteria, data extraction, and data analysis procedure.

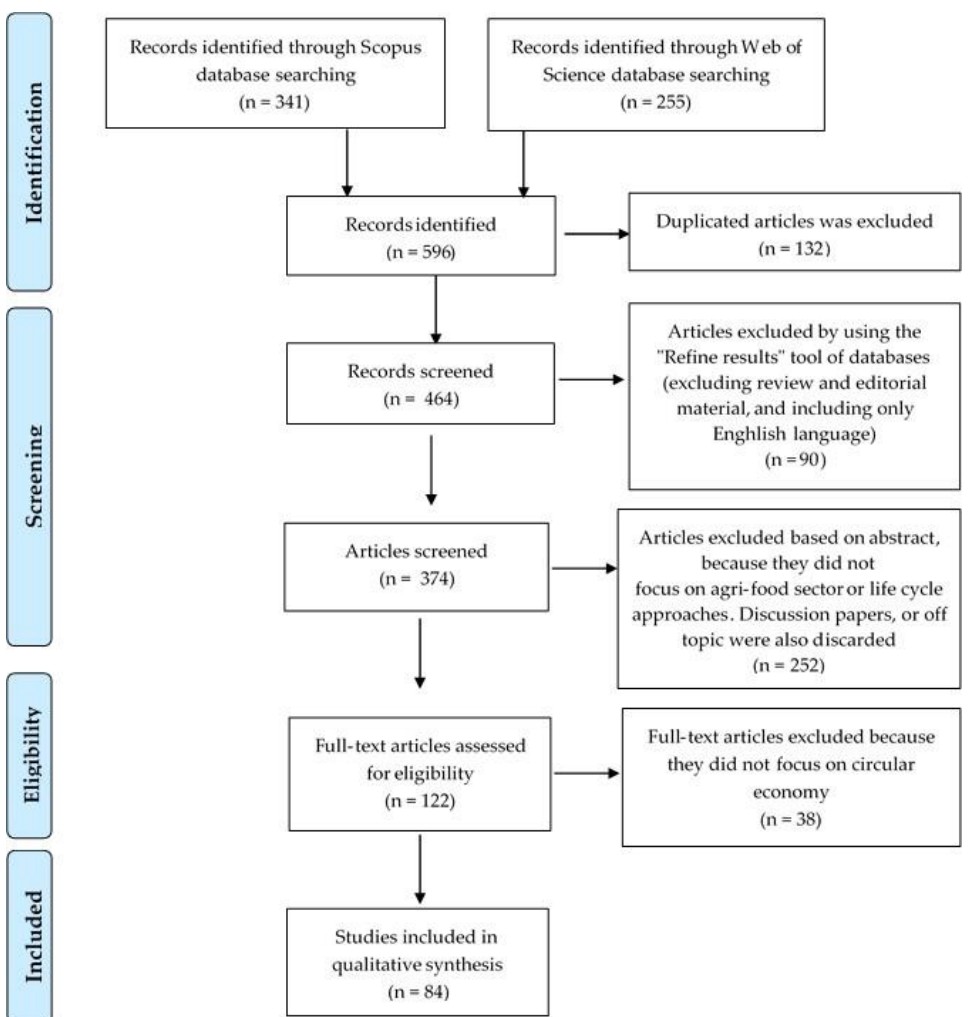

**Figure 2.** Methodological steps of the literature search process using the Preferred Reporting Items for Systematic Reviews and Meta-Analyses (PRISMA) flow diagram [28].

In the "identification" step of the PRISMA flow diagram (cf. Figure 2), a set of keywords was selected based on the question formulation, i.e., the research scope, which consisted of searching for all documents proposing life cycle approaches to measure the empirical circular pathways of agri-food systems. The literature search was performed in Scopus and Web of Science (WOS) databases, through the combination of main keywords using Boolean operators (AND/OR). As shown in Table 1, the following search strings were applied: ("circular economy"), ("life cycle assessment" OR "life cycle analysis" OR LCA),

("life cycle costing" OR LCC), ("social life cycle assessment" OR "S-LCA" OR SLCA OR "social-LCA"), ("life cycle sustainability assessment" OR LCSA) combined with ("agr*" OR food). The research has been conducted in the fields "title", "abstract", and "keywords" for the main keywords, and in "all fields" for the other terms, i.e., agr* or food. The databases were consulted in October 2020 with no time restriction.

**Table 1.** Query used in database searching.

| Database | Search Strings [1] |
| --- | --- |
| Scopus | (TITLE-ABS-KEY ("circular economy") AND TITLE-ABS-KEY ("life cycle assessment" OR "life cycle analysis" OR "life cycle costing" OR "social life cycle assessment" OR "life cycle sustainability assessment" OR LCA OR LCC OR "S-LCA" OR SLCA OR "social-LCA" OR LCSA) AND ALL (agr* OR food)) |
| Web of Science | TOPIC: ("circular economy") AND TOPIC: ("life cycle assessment" OR "life cycle analysis" OR "life cycle costing" OR "social life cycle assessment" OR "life cycle sustainability assessment" OR LCA OR LCC OR S-LCA OR SLCA OR "social-LCA" OR LCSA) AND ALL FIELDS: (agr* OR food) |

[1] Last accessed on 29 October 2020.

Searches on Scopus and WOS databases led to 341 and 255 articles, respectively, for a total of 596 papers. Duplicate papers were excluded, resulting in 464 documents, which have been subjected to a screening process. A first selection was made by using the "Refine Results" tool of the databases used to exclude review and editorial material and include only the English language. Then, only applicative indexed references were taken into consideration. A second screening was performed based on the content of abstracts, excluding discussion papers, or off-topic and studies that did not focus on the agri-food sector or life cycle approaches. In so doing, 122 articles were assessed for eligibility by reading the full-text in-depth. Studies not directly focused on the issue of measuring circularity quantitatively were discarded.

Through the above-specified criteria application, the total amount of articles found was reduced to a final portfolio of 84 representative papers that were included in the qualitative synthesis. These articles were read in full and analyzed one by one for the purpose of this study.

### 2.2. Characterization of Matrix Criteria for the Systematic and Critical Review

According to De Luca et al. [29], an evaluation matrix has been set up to synthesize research evidence, by classifying the selected papers according to several integrated criteria. As shown in Table 2, all reviewed papers have been categorized by bibliometric information (authors, year of issue, title, journal); descriptive statistics that refers to the place where the case-study is applied; field of application (i.e., the area of human activity); the main product under study; circularity topics; and relevant data on circularity assessment methods and circularity indicators. These latter criteria included the differentiation of both the methodologies into "LC tools and other life cycle approaches" and "other methods different from LC", and indicators into "circularity indices" and "CE assessment indicators". According to Corona et al. [30], the former indicators measure the circularity degree of a system, based on a mere material recirculation, and address resource efficiency. The latter assesses the effects (burden or value) of circularity, showing high potential in addressing all the CE goals at the product/service level. Here, we divided the "CE assessment indicators" into "lifecycle-based indicators" and "not life cycle-based indicators" (see Section 3.3 for more details). "Circular strategy application-level" indicates the levels to which the circular strategies or interventions are applied, namely, micro-level (e.g., products, companies or organizations, consumers), meso-level (e.g., eco-industrial parks), and macro-level (e.g., regions, cities, countries, or the global economy) [31]. Finally, the last columns of the matrix are focused on the main features that qualify the life cycle approaches, e.g., functional unit, system boundary, database, LC impact assessment method, software, etc. (see Table S1).

Once the matrix has been completed, the input data were compared and the results were qualitatively and quantitatively extracted to highlight significant information and

relationships. The main highlights and conclusions of the selected studies are reported in the following section.

**Table 2.** Matrix criteria for the critical review of the selected papers.

| Criteria | Description |
|---|---|
| ID Paper, Authors, Year, Title, Source | Bibliometric information. The sequence follows the alphabetical order of the first author's name |
| Place | Where the case-study took place |
| Field of application | What is the context in which the application is implemented |
| Main reference product | What is the product analyzed in the case-study |
| Circularity topics | The most common arguments leading the CE literature |
| Circularity assessment methods | LC tools and other life cycle approaches/Other methods different from LC |
| Circularity indicators | Circularity indices (measuring the circular degree of a system) |
| | CE assessment indicators (assessing the effects of circularity) divided into life cycle-based indicators and not life cycle-based indicators |
| Circular strategy application level | Macro, Meso, Micro |
| LC approach details | Functional unit, System boundary, Data/Database, LC impact assessment method and/or software, Type of cost, Approach used |

## 3. Results

### 3.1. Descriptive Analysis

The descriptive analysis was based on the distribution of the reviewed articles over the years and by country (based on the place of case-study application), and their distribution per journal, field of application (resulted from the main argument or topic of study), main reference product (which refers to the product analyzed in the case-study), and the most common topics dominating the CE literature in the agri-food sector.

The selected 84 papers were published from 2014 to 2021, as shown in Figure 3. It should be noted that the papers issued in 2021 were already available online in October 2020. The results revealed an exponential increase in the number of publications regarding the application of life cycle methodologies as circularity metrics in the agri-food sector over the last seven years. The first publications, starting from 2014 to 2016 (one for each year), concerning energy recovery from dairy farming [32], recycling food waste for use as feed in aquaculture [33], and the waste management with energy recovery in the anchovy industry [34]. Only in recent years, more specifically in 2018–2020, strong efforts were made toward the development of studies to measure the circularity through LC approaches in agri-food systems. In particular, 16 documents were published in 2018, 23 documents in 2019, and 32 documents in 2020. Nevertheless, as mentioned by Vetroni Barros et al. [17], only a few efforts can be observed toward assessing agricultural systems accounting for sustainability in a circular perspective. Thus, the peak development of this theme has yet to be reached.

According to the place of case-study application, most publications may be traced to European countries (49%), followed by China, as shown in Figure 4. Indeed, the five highest-ranked origins of the reviewed articles are Spain (22.6%), Italy (14.3), United Kingdom (UK) (7.1%), China (6%), and Ireland (4.8%). These findings were consistent with results reported by Esposito et al. [26], who showed the great interest of European scholars toward the development of CE models also in the agri-food sector. Outside the European continent, China represents the major contributor in researching this topic. This is likely due to the Chinese government's request to stimulate actions in favor of the environment, also via CE [35]. The interest in using LC tools to analyze CE strategy seems to be growing in Brazil and Sweden with three publications each (for more information, refer to Table S1 in the Supplementary Materials Section).

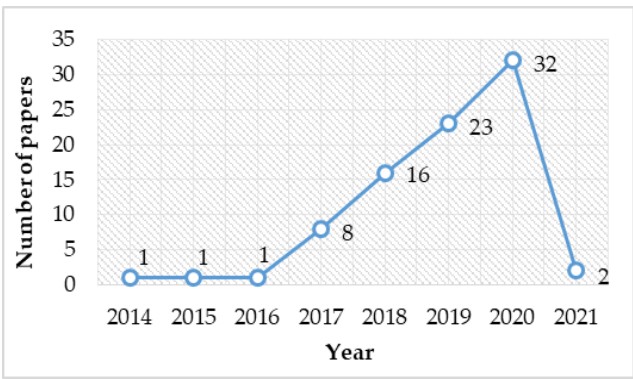

**Figure 3.** Publication trend by year. Paper search ended on 29 October 2020.

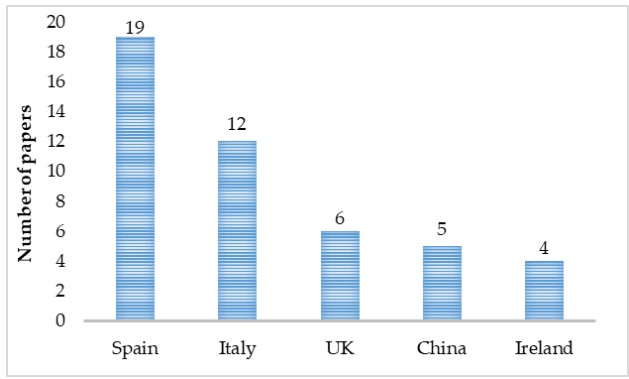

**Figure 4.** Geographical distribution by top publishing country.

Concerning the type of contribution, 83 articles out of 84 were published in scientific peer-reviewed journals and only one in proceedings of scientific international conferences. The highest-ranked journals were *Journal of Cleaner Production* (21), *Science of the Total Environment* (10), *Resources, Conservation and Recycling* (8), *Waste Management* (5), and *Sustainability* (4), with 57.1% of documents considered. This is due to the scope of these journals also related to the theme of the CE. The remaining journals showcased one or two publications each. All scientific journals addressed sustainability topics and environmental issues, only one specialized in agricultural systems and food production (*Agricultural Systems*).

Figure 5 presents the main argument covered in the studies analyzed. In this review, we refer to the area of human activity or context in which the application is implemented (Table S2). The waste and/or biomass fields of application were the most addressed by the published articles accounting for 55% of the total, of which 24 documents (29%) are strictly dedicated to wastes, 20 (24%) to biomass, and 2 (2%) to the whole of "wastes and biomass". Here, "wastes" refers to the use and recycling of household wastes [36], wastewater [37], agricultural wastes [38,39], food waste [40–42], and organic waste [43,44], including the recovery of nutrients [45], organic compounds, and energy [46]. This field is of great interest to European societies and academics, and constantly under the spotlight due to the recent publication of the waste management directives [47] that fall within the "Circular Economy Action Plan" adopted by the European Commission in December 2015 [48]. In the field named "biomass", several kinds of goods (wood, garbage, crops, fruits, litter, manure, landfills gas, etc.) for energetic purposes (anaerobic digestion [49–53], etc.) were included.

Around 15% of the studies were included in the "manufacturing" field, which includes product production from raw (renewable or non-renewable) materials. For instance, the manufacture of biochemicals and bio-based plastics is one of the strategies promoted by the European Union within the Europe 2020 strategy [54], and the production of bio-based packaging is an effective and promising climate change mitigation strategy [55].

The "agriculture" field, accounting for 11% of the total, enclosed production of fruits and vegetables [56,57] for fresh consumption or industrial transformation [58] (raw materials, food, and no food).

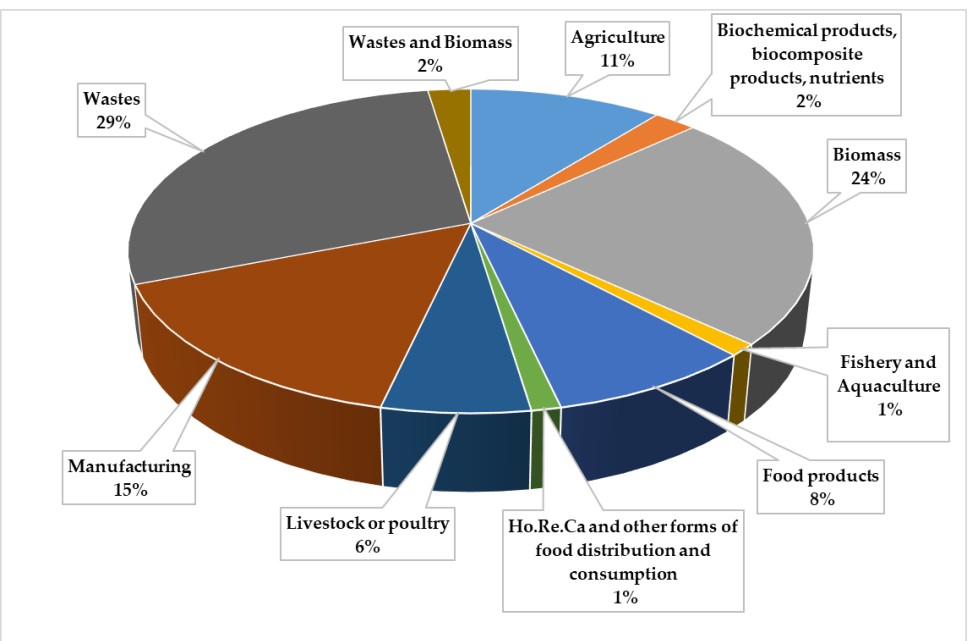

**Figure 5.** Fields of application.

Considering the reference product analyzed in the case studies, Figure 6 shows the most common ones in the selected papers. With 11% of the total, "food waste" is the most represented product category. In this group, the authors were considered a heterogeneous set of products [36,40–42]. For instance, Cristóbal et al. [41] attempted to identify the optimal combination of food waste prevention by analyzing the quantity of food waste generated along five different food supply chains (i.e., grain, meat, fruit and vegetable produce, milk/dairy, and seafood). De Sadeleer et al. [36] analyzed the avoidable food waste amounts contained in household waste (fruits and vegetables, bread and pastries, fish, meat, dairy products, eggs, meal leftovers), for waste prevention, energy recovery, or recycling purposes.

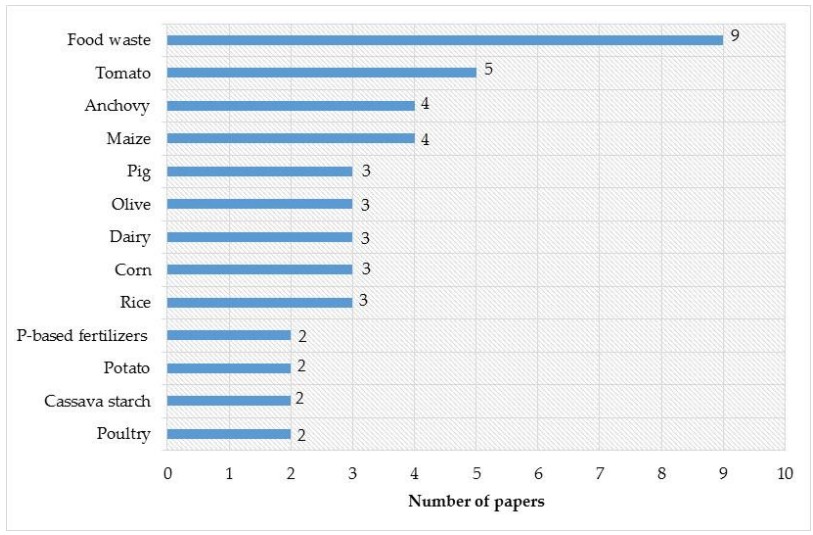

**Figure 6.** The main reference products in the literature review.

Other main reference products identified in this review included several agro-industrial products, such as tomato [58–62], anchovy [34,46,63,64], maize [52,57,65], pig [66–68], olive [69–71], dairy [32,72,73], corn [43,45,74], rice [56,75,76], potato [77,78], poultry [49,79], cassava starch [51,55], beer [80], and coffee [81].

The need to recycle nutrients such as phosphorus has been widely considered as an important issue of a CE. For instance, Svanström [82] and Smol [83] carried out an environmental evaluation of technologies for phosphorus recovery from sewage sludge to be applied on agricultural land as fertilizers (P-based fertilizers). As argued by the authors, phosphorus in wastewater should be utilized to avoid depletion of mineral phosphorus reserves, in line with the principles of a CE.

The most common "circularity topics" that emerged in this study's final portfolio were closed-loop production systems, e.g., nutrient recovery for agricultural purposes, production of renewable energy, valorization of residues and wastes, food waste, and agro-wastes recycling for agriculture, in addition to the reduction of input, final wastes, or product losses (Figure 7, Table S3). Here, the topic of "waste valorization", accounting for 32% of the total, is used to indicate retrieve elements from wastes or losses to be used for new purposes, such as extraction of biochemical feedstock and nutrients recovery [84–88]. The second main circularity topic issued in this study refers to energy recovery, with about 29% of the final portfolio. Incineration of material, usually biomass, with energy recovery [89], composting for energetic purposes [90], and anaerobic digestion [72,78,91] were the recurring questions include in this topic.

Moreover, the "recycle" topic was observed in 15% of the total documents. In the present review, we refer to turning an item (products, co-products, by-products) into raw materials that can be used again, usually for a completely new product. For instance, composting and packaging recycling [92–94]. In this topic, energetic purposes are excluded.

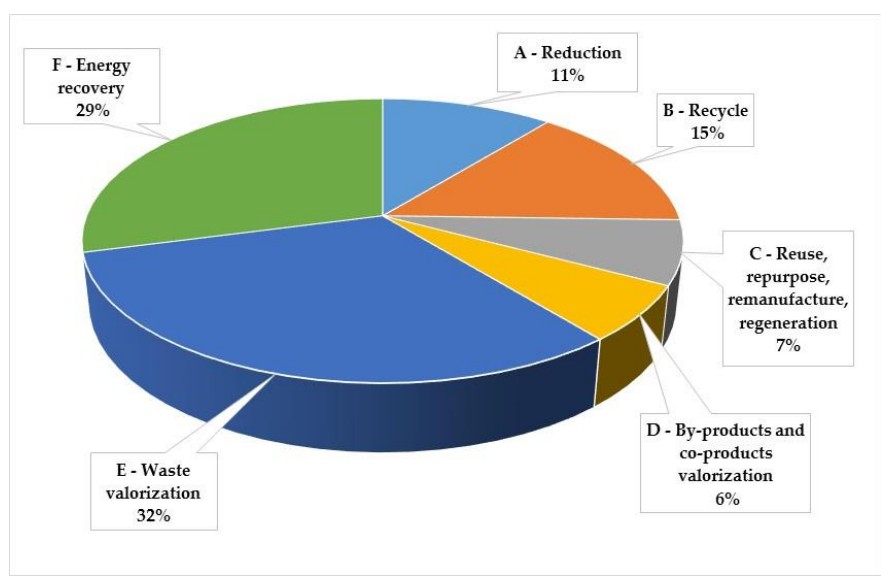

**Figure 7.** Most common "circularity topics" in the literature review.

*3.2. Circularity Assessment Methods Based on LC Tools and Other Life Cycle Approaches*

According to the findings shown in Figure 8, the most common LC tool adopted by papers to assess the benefits/impacts of CE strategies is LCA. This review found 52 case studies out of 84 (62% of the total) using stand-alone LCA. LCA is considered by all authors as the most suitable methodology to assess products, services, technologies in a CE perspective, including studies on biomass for energetic purposes, food products, biochemical and bio-composite products, waste reduction and waste valorization also for energy recovery, and manufacturing of products from raw (renewable or non-renewable)

materials. Most of the papers were published in 2019 and 2020, indicating how LCA in the agri-food sector toward CE is a quite recent topic of research.

In contrast, only eight studies (9.5%) deal with the LCC methodology combined with other analyses. Of these, six adopt LCC combined with LCA [40,43,46,54,73,95], one paper combined the LCC model with LCA and material flow analysis (MFA) [75], and another with externality analysis in the CE perspective [96]. Finally, no paper dealt with the S-LCA methodology.

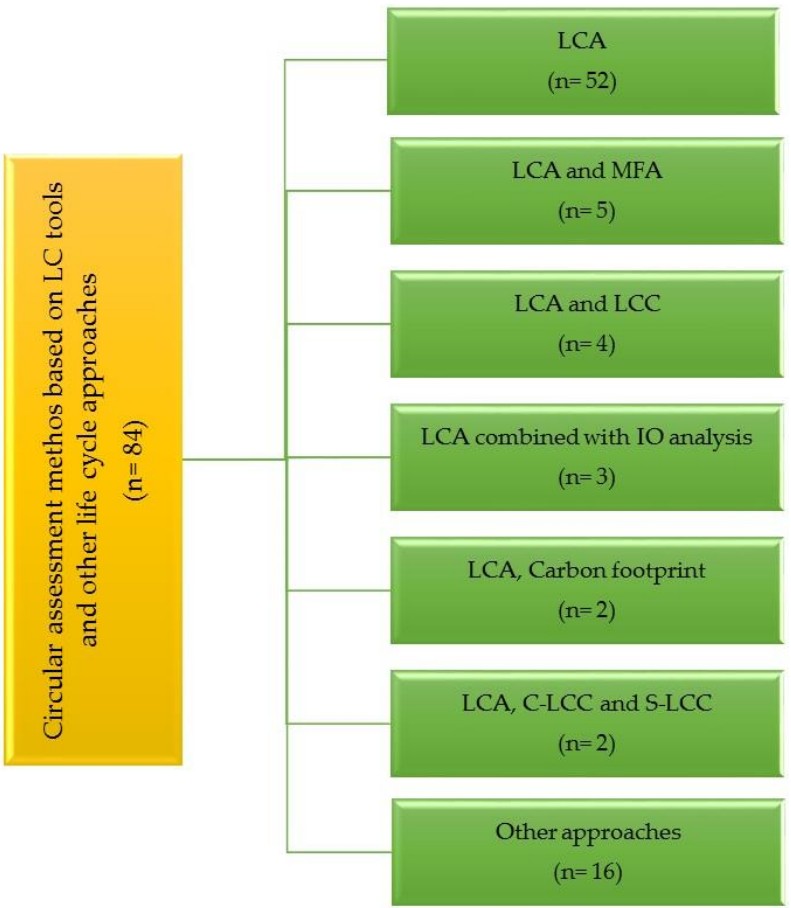

**Figure 8.** The main circularity assessment methods based on life cycle (LC) tools and life cycle approaches in the literature review. (LCA = life cycle assessment, LCC = life cycle costing, MFA = material flow analysis, IO = input–output analysis, C-LCC = conventional life cycle Costing, S-LCC = societal-life cycle costing, other approaches = LC inventory analysis, mass and energy balances, cumulative energy demand, energy flow analysis, life cycle protein assessment (LCPA), LCA-based waste footprint metric, material flow model, eco-efficiency analysis, Emergy accounting method (EMA)).

As described in Table 3, this literature review found 52 case studies applying stand-alone LCA to analyze the contribution of circular strategies to the principle of CE. Most of the reviewed LCA is performed following several impact evaluation methods that include multiple indicators representing up to 16 different impact categories.

In the final portfolio of papers, the most common LCA indicators were "global warming potential" (or "climate change" or "carbon footprint") applied in 58 papers (67% of the total), "eutrophication" (for marine, freshwater, and terrestrial ecosystems) in 45 papers (55%), "human toxicity" in 28 papers (35%), and "ecotoxicity" in 25 papers (30%). According to Berti et al. [97], these impact categories have been documented to be the most appropriate for agricultural assessments.

The most applied method was "ReCiPe", accounting for 38.5% of the total papers. For instance, Beausang et al. [49], used the ReCiPe method in the hierarchic perspective to conduct a consequential LCA to examine several scenarios in which biogas produced from poultry litter is used to generate heat and electricity or is upgraded to biomethane, which can substitute natural gas. Among the impact categories (midpoint level) selected by the authors, climate change, acidification, and eutrophication are recommended impact categories for LCA of bioenergy systems. Moreover, these impact categories related to carbon, nitrogen, and phosphorous flows are significant for agricultural systems. The ReCiPe midpoint method was chosen by Buonocore et al. [50], allowing for the assessment of the contribution of wood-based bioenergy plant, which utilizes local residues from wood industries and forestry operations, through several impact categories, i.e., climate change (GWP), fossil depletion (FD), ecotoxicity (FEP), human toxicity (HTP), photochemical oxidant formation (POFP), etc. Corcelli et al. [59] and Cortés et al. [38] also used the ReCiPe method in a hierarchic perspective and at the midpoint level. In the first case study, the authors conducted a life cycle assessment of different productive uses of rooftops under Mediterranean climatic conditions, while, in the second case study, an evaluation of the environmental burdens of composting as a way to achieve a more circular valorization of wine waste.

The other two most applied methods in the literature review were CML (Centrum voor Milieuwetenschappen in Leiden) (21.2%) and ILCD (International Reference Life Cycle Data System) (17.3%). Campos et al. [79], for example, assessed four environmental impact categories using the CML method, i.e., global warming (GW), abiotic depletion (AD), acidification (AC), and eutrophication (EUT), to carry out an environmental life cycle assessment of poultry fat, poultry by-product meal and steam hydrolyzed feather meal obtained by rendering poultry byproducts. The CML impact assessment method and eight of its impact categories (global warming potential (GWP100a), human toxicity (HT), freshwater aquatic ecotoxicity (FW), marine aquatic ecotoxicity (MAE), terrestrial ecotoxicity (TE), photochemical oxidation (PO), acidification (AC) and eutrophication (EUT)), were applied by Krishnan et al. [98] for assessing the environmental impact of a redesigned mango food supply chain to improve environmental sustainability.

To evaluate the environmental performance of the animal feed from *Camelina sativa*, Martinez et al. [99] used ILCD 2011 midpoint method, and also considered the following impact categories: climate change, human toxicity, terrestrial acidification, freshwater eutrophication, terrestrial eutrophication, and abiotic depletion. ILCD midpoint characterization method was also adopted by Tedesco et al. [100] to evaluate the environmental impact of the bioconversion of fruit and vegetable waste into earthworm meal to be used as a new food/feed source.

Considering the importance of energy consumption in the agricultural systems, some authors also included in their analyses the cumulative energy demand (CED), an impact indicator that expresses the energy consumption throughout the life cycle of a product or a service [33,51,61,65,93,101,102]. Others focused on the primary energy demand (PED), which represents an appropriate indicator for illustrating the interactions of the food-energy nexus [58,60,81,91,103,104].

**Table 3.** LCA framework in the literature review.

| | Authors | Case Studies | LCA Application | Impact Evaluation Method/Categories or Indicators * |
|---|---|---|---|---|
| 1 | Beausang et al. 2020 [49] | Poultry | Consequential LCA | ReCiPe 2016 method (midpoint level) |
| 2 | Belaud et al. 2019 [75] | Rice | LCA | ReCiPe 2016 method (midpoint and endpoint level) |
| 3 | Boesen et al. 2019 [92] | Liquid food | Streamlined LCA | ILCD 2011 method |
| 4 | Buonocore et al. 2019 [50] | Wood | LCA | ReCiPe method (midpoint level) |
| 5 | Campos et al. 2020 [79] | Poultry | LCA | CML method |
| 6 | Cascone et al. 2020 [93] | Plastic films | LCA | ReCiPe (endpoint level) + IPCC 2013 GWP 100a + CED + WFA |

**Table 3.** *Cont.*

| | Authors | Case Studies | LCA Application | Impact Evaluation Method/Categories or Indicators * |
|---|---|---|---|---|
| 7 | Casson et al. 2020 [105] | Street food | LCA | ILCD 2011 midpoint method |
| 8 | Chaudron et al. 2019 [84] | Cranberry juice | LCA | IMPACT 2002 + method |
| 9 | Colley et al. 2020 [20] | Meat | LCA | ReCiPe (endpoint level) + CML + TRACI + USETOX methods |
| 10 | Corcelli et al. 2019 [59] | Tomato | LCA | ReCiPe Midpoint method |
| 11 | Cortés et al. 2020 [38] | Viticulture | LCA | ReCiPe 2016 method (midpoint level) |
| 12 | Cristóbal et al. 2018 [41] | Food waste | LCA | ILCD method |
| 13 | Eggemann et al. 2020 [53] | Cattle manure and straw residues | Attributional LCA | ReCiPe 2016 method (midpoint level) |
| 14 | Gaglio et al. 2019 [65] | Maize-germ oil | LCA | CML-IA baseline method |
| 15 | Keng et al. 2020 [94] | Food waste | LCA | TRACI 2.0 method |
| 16 | Krishnan et al. 2020 [98] | Mango | LCA | CML-IA method + WFA |
| 17 | Lansche et al. 2020 [51] | Cassava starch | LCA | CED + DEF + WSI + GWP + OFP + AP + HTP + ETP |
| 18 | Laso et al. 2016 [34] | Anchovy | LCA | IChemE, 2002 metrics + AA + GW + HHE + SOD + POF + AqA + AOD + Meco + NMEco + EU |
| 19 | Laso et al. 2018b [63] | Anchovy | LCA | Wc + Ec + Pc + Cei + EROI method |
| 20 | Liu et al. 2018 [106] | Corn straw | LCA | Life-cycle fossil primary energy and GHG emissions |
| 21 | Lokesh et al. 2020 [107] | Corn and sugar-beet | LCA | IPCC GWP + UNEP model + Accumulated exceedance model + EUTREND model—ReCIPe 2008 + USEtox model + CML 2002 + AWARE methods |
| 22 | Lucchetti et al. 2019 [85] | Ecological detergent | Partial LCA | EcoIndicator 99 method |
| 23 | Martin et al. 2019 [86] | Brewers' spent grains | LCA | CML 2014 method |
| 24 | Martinez et al. 2020 [98] | *Camelina sativa* | LCA | ILCD 2011 Midpoint + |
| 25 | Monsiváis-Alonso et al. 2020 [44] | Fish oil | LCA | ReCiPe 2016 method |
| 26 | Niero et al. 2019 [80] | Beer | LCA | ILCD method |
| 27 | Noya et al. 2017 [66] | Pig | LCA | ReCiPe Midpoint method |
| 28 | Oldfield et al. 2017 [58] | Tomato | LCA | CML 2001 method + PED |
| 29 | Oldfield et al. 2018 [108] | Food waste | LCA | CML midpoint method |
| 30 | Pérez-Camacho et al. 2018 [109] | Maize/grass silage and cattle manure | LCA | ReCiPe method (midpoint level) |
| 31 | Piezer et al. 2019 [60] | Tomato | LCA | PED + fossil fuels + renewable energy + dissipation |
| 32 | Qin et al. 2018 [39] | Tobacco | LCA | TRACI method (midpoint level) |
| 33 | Roffeis et al. 2017 [87] | *Musca domestica* and *Hermetia illucens* | LCA | not explicit |
| 34 | Roffeis et al. 2020 [88] | *Musca domestica* and *Hermetia illucens* | Attributional LCA | ReCiPe method (midpoint and endpoint level) |
| 35 | Rufí-Salís et al. 2020a [61] | Struvite recovery | LCA | ReCiPe 2016 method (midpoint level) + CED |
| 36 | Rufí-Salís et al. 2020b [110] | Tomato | LCA | ReCiPe 2016 Midpoint method |
| 37 | Rufí-Salís et al. 2020c [111] | Green bean | LCA | ReCiPe Midpoint method |
| 38 | Santagata et al. 2017 [112] | Animal waste | LCA | ReCiPe Midpoint method |
| 39 | Santiago et al. 2020 [100] | Onion | LCA | CML 2001 method + CED |
| 40 | Schmidt Rivera et al. 2020 [105] | Coffee | LCA | Recipe 2016 method + PED |
| 41 | Schmidt Rivera et al. 2019 [81] | Raspberries and meat | LCA | Climate change, Depletion of fossil fuels, Depletion of metals + PED |
| 42 | Sierra-Perez et al. 2018 [101] | Cork | LCA | ReCiPe 2008 method + CED |

Table 3. *Cont.*

|  | Authors | Case Studies | LCA Application | Impact Evaluation Method/Categories or Indicators * |
|---|---|---|---|---|
| 43 | Slorach et al. 2019a [91] | Food waste | LCA | ReCiPe method (midpoin level) + PED |
| 44 | Slorach et al. 2019b [103] | Food waste | LCA | ReCiPe + Thinkstep (PED) methods |
| 45 | Smol et al. 2020 [83] | P-based fertilizers | LCA | ILCD 2011 Midpoint + method |
| 46 | Strazza et al. 2015 [33] | Salmon breeding | LCA | IPCC report (GWP) + CED + WSI |
| 47 | Svanström et al. 2017 [82] | P-based fertilizers | LCA | ILCD method |
| 48 | Tedesco et al. 2019 [99] | Earthworm meal | Attributional LCA | ILCD midpoint method |
| 49 | Uceda-Rodríguezet al. 2020 [71] | Olive | LCA | CML 2000 (midpoint level) |
| 50 | Vaneeckhauete al. 2018 [67] | Pig | LCA | CML 2010 |
| 51 | Wohner et al. 2019 [73] | Dairy | Streamlined LCA | ILCD 2018 method |
| 52 | Wolsey et al. 2018 [89] | Willow biomass | LCA | not explicit |

* (CED = cumulated energy demand, DEF = deforestation, WSI = water stress index, WFA = water footprint assessment, PED = primary energy demand, GWP = global warming potential, OFP = photochemical ozone formation potential, AP = acidification potential, HTP = human toxicity potential, ETP = aquatic ecotoxicity potential, AA = atmospheric acidification, GW = global warming, HHE = human health (carcinogenic) effects, SOD = stratospheric ozone depletion, POF = photochemical ozone (smog) formation, AqA = aquatic acidification, AOD = aquatic oxygen demand, MEco = ecotoxicity to aquatic life (metals to seawater), NMEco = ecotoxicity to aquatic life (other substances, EU = eutrophication. EROI = energy return on investment, Wc = water consumption, Ec = energy consumption, Pc = food, Cei = climate).

Few studies used water footprint (WF) indicator [93,105], known worldwide for the assessment of environmental performance. Among the methods involved in LCA-based water footprint, the authors adopted the water footprint assessment (WFA), a method centered upon computation of the water stress index (WSI) that calculates the water impact on the consumption-to-availability perspective of freshwater deprivation.

As above-mentioned, few studies adopted the LCC methodology as a tool for measuring CE strategies from an economic point of view. Table 4 summarizes all the reviewed papers that implemented LCC combined with other approaches. Albizzati et al. [40] and Blanc et al. [54] performed a conventional (C-LCC) and societal (S-LCC) life cycle costing paired with LCA. The first authors used LCC to evaluate the socio-economic impacts of producing wet animal feed, protein-concentrated animal feed, and lactic, polylactic, and succinic acid from food waste. The LCC model implemented was the unit-cost method approach, where the waste management system under study is divided into stages (e.g., collection, transport, processing) and each stage is characterized by its relevant costs, classified into budgets costs, transfers, and externalities. As argued by the scholars, LCC provides critical insights into process performance, giving a platform for more targeted technology optimization. The second authors used C-LCC and S-LCC to assess the economic aspects of the use of bio-based plastics in the fruit chain along the whole chain, following the methodological scheme expressed by Gluch and Baumann [113] and Neugebauer et al. [114]. Environmental externalities and their relative monetary value were also identified.

By combining the LCC model and externalities in the CE, [96] analyzed the benefits of using aluminum packaging in the food sector. The approach proposed by [115] was used to evaluate costs and benefit and to externalities. As discussed by the researchers, it is necessary to adopt the LCC approach as a useful economic model to guide the solutions for sustainable manufacturing and the CE vision. Cobo et al. [74] and Cobo et al. [43] used the economic model derived from the solid waste optimization life cycle framework (SWOLF) to perform an LCC analysis to evaluate a waste management system that aims at recovering nutrients from municipal organic waste. Laso et al. [46] performed an LCC analysis based on the approaches described by Hunkeler et al. [115] and Swarr et al. [116] to assess the costs related to different waste management alternatives from the fish canning industry. The scholars suggest that LCC can help to identify all steps that constitute an opportunity to reduce costs, helping decision-makers to choose a cost-effective project alternative. To estimate the economic implications of recovering energy and material resources from food waste, Slorach et al. [91] applied LCC methodology following the guidelines published by

Swarr et al. [116] and Hunkeler et al. [115]. Finally, Wohner et al. [62] used LCC taking the value-added approach (VA) to evaluate the economic aspects of packaging-related food loss and waste of food-packaging systems.

**Table 4.** Main LCC features in the reviewed papers.

| | Authors | Case Studies | LCC Framework | LCC Features | | |
|---|---|---|---|---|---|---|
| | | | | Approach Used | Type of Costs | Data |
| 1 | Albizzati et al. (2021) [40] | Food waste | LCA, C-LCC, and S-LCC | Unit-cost method approach | - Budgets costs, transfers, and externalities | Statistic, Literature |
| 2 | Albuquerque et al. (2019) [96] | Aluminum and tinplate | LCC and PSILA, Externalities | Approach proposed by Hunkeler et al. (2008) | - Production cost<br>- Overhead Costs<br>- Depreciation | Interviews to stakeholder |
| 3 | Blanc et al. (2019) [54] | Berry fruit | LCA, C-LCC, and S-LCC | Methodological scheme expressed by Gluch and Baumann, 2004, and Neugebauer et al. (2016) | - Conventional costs for agricultural operations<br>- Costs incurred for product transformation, sales, consumption, and disposal of waste<br>- Environmental externalities | Face-to-face interviews with different actors |
| 4 | Cobo et al. 2020 [43] | Corn and wheat | LCA and LCC | Economic model derived from SWOLF | - Capital costs of the unit processes<br>- Costs associated with the farmers' equipment and land | Data from SWOLF |
| 5 | Cobo et al. 2019 [74] | Corn | MFA, LCA, and LCC | Economic model derived from SWOLF | - Waste management costs | Data from SWOLF, Literature |
| 6 | Laso et al. 2018a [46] | Anchovy | LCA and LCC | Approaches described by Hunkeler et al. (2008) and Swarr et al. (2011) | - Costs of raw materials<br>- Costs of anchovy processing and manufacturing<br>- Costs of packaging<br>- Management costs of waste treatment<br>- Value Added | Literature, Market reports, and actor information |
| 7 | Slorach et al. 2019c [95] | Food waste | LCA and LCC | Guidelines published by Swarr et al. (2011) and Hunkeler et al. (2008) | - Costs to local authorities, operators of treatment facilities, and consumers | Literature, Statistics |
| 8 | Wohner et al. 2020 [62] | Tomato | Streamlined LCA and LCC | Value-added approach (VA) | - Costs to the ketchup producer for purchasing ingredients, energy, and packaging<br>- VA of agricultural production of ingredients, energy and packaging, transports | Ecoinvent 3.5 database |

(LCA = life cycle assessment, LCC = life cycle costing, MFA = material flow analysis, C-LCC = conventional life cycle costing, S-LCC = societal-life cycle costing, SWOLF = solid waste optimization lifecycle framework, VA = value-added).

In many case studies reviewed, CE strategies were assessed through LCA combined with other "life cycle-type" approaches, i.e., methods not directly ascribed to typical LC framework (i.e, LCA, LCC, and S-LCA), but that approached the evaluation process in a life cycle perspective. Among the other methodological approaches most applied, there were material flow analysis (MFA), input–output (IO) analysis, and carbon footprint, implemented coherently with principles and methodological steps of an LC-based approach.

Five case studies adopted MFA accounting combined with LCA to evaluate the circularity of systems. Following the MFA modeling principles of Brunner and Rechberger [117],

material flows of a system are measured in terms of their mass. De Sadeleer et al. [36] compared the environmental benefits of household food waste prevention to the benefits from various waste management strategies concerning recycling rates, energy efficiency, and emission efficiency, by using the MFA model combined with published LCA results. The authors suggest that the most effective food waste management strategy seems to be a combination of prevention and recycling strategies. However, for mitigating climate change, the prevention of food waste clearly stood out as the most effective strategy. Cobo et al. [118] studied the optimal configuration of a waste management system that valorizes the municipal organic waste (OW) in Cantabria, performing an MFA of the system and an LCA of the unit processes concerning the treatment of solid OW and the land application subsystem. The closed-loop perspective of the system analyzed by the authors is given by the application of products generated from the OW (compost, digestate, etc.) to land, which results in a reduction in the consumption of the industrial fertilizers. Their results indicated that an enhanced circularity of resources does not necessarily entail the decrease of both the overall consumption of natural resources and the emission of environmental burdens of the system. Cobo et al. [45] also considered the circularity of nutrients within a system that handles the organic waste generated in Cantabria. They concluded that improving nutrient circularity paradoxically leads to eutrophication impacts, and increasing the source separation rates of OW has a positive effect on the carbon footprint of the system. Hadin et al. [90] developed an LCA approach based on MFA to calculate the potential environmental impact of combined energy recovery and nutrient recycling from horse manure through anaerobic digestion in a centralized plant, replacing unmanaged composting. The authors indicated that anaerobic digestion is suitable to reduce potential environmental impact in comparison to unmanaged composting, mainly due to biogas substituting the use of fossil fuels. Finally, Stanchev et al. [72] proposed an approach based on MFA and LCA for measuring the environmental performance of the anaerobic treatment of dairy processing effluents based on the CE principles. Their results showed that the recovered energy from AD provides 20% of the energy requirements of the factory reducing the total carbon footprint emissions by 13% compared to the baseline scenario.

This analysis found three studies applying input–output (IO) technique combined with LCA. According to Miller and Blair [119], IO analysis uses a top-down, economic method to capture product and service flow from one industrial sector to all other sectors within one country, region, or multi-regions. As argued by Corona et al. [30], this type of top-down approach has been applied by the LCA community to compensate for the shortcomings of process-based LCA (e.g., expanding the scope from the product level to national/global level). Chen et al. [76] and Chen et al. [77], developed a hybrid life cycle assessment model integrating process-based LCA with IO analysis. In the first study, the authors implemented such an approach to evaluate the feasibility and potential benefits of a novel bio-fertilizer technology that utilized paddy rice residues through composting. The bio-fertilizer can recycle the nutrients in residues to replace synthetic fertilizer within a circular rice production system. In the second study, the authors used a hybrid IO–LCA model to assess the environmental, social, and economic impacts of modifying conventional bioplastics production with a potato pulp residue leftover from starch production to produce biocomposite. Ruiz-Peñalver et al. [120] also applied an IO-based hybrid LCA model to estimate total waste generation throughout the supply chain in Spain.

Only two studies applied the Carbon footprint of a product by using the LCA approach. Arunrat et al. [56] used the LCA concept for greenhouse gas emissions (LCA–GHG) to evaluate and compare GHG emissions of large-scale and individual farming in rice production, while Xue et al. [68] used LCA for analyzing carbon footprints in traditional and biogas-based circular economic models of pig farming.

### 3.3. Reviewed Circularity Assessment Indicators

As shown in Table 5, only eight articles deal with the CE assessment through specific indicators. As previously stated, in this review we refer to the classification of CE assess-

ment indicators proposed by [30], who identify indicators that measure the circularity degree of a system, based on a mere material recirculation and addressed to resource efficiency, and indicators that assess the effects (burden or value) of circularity. Here, by life cycle-based indicators, we refer to the life cycle impact categories indicators retrieved from LCA, the LCC indicators when utilized for evaluating CE strategies, and stand-alone indicators based on life cycle approaches.

Purposes and advantages derived by the use of CE indicators have been widely argued in the literature [31,121–123]. Due to the complexity inborn in the circularity economic paradigm, CE indicators combining different metrics can deliver simplified results.

Some CE indicators examined in this review were developed to assess the circular degree of a system. Within this topic, Cobo et al. [45] developed indicators to study the circularity of nutrients within a system that handles organic waste (OW) generated in Spain. More in detail, the circularity indicators of carbon (CIC), nitrogen (CIN), and phosphorus (CIP) have been applied to a circular integrated waste management system, which encompasses waste management and the processing and consumption of the components recovered from waste and the external raw materials. As argued by the authors, enhancing the circularity of these nutrients seems to be a suitable strategy for closing their natural biogeochemical cycles by avoiding the accumulation of nutrients in one of the earth's subsystems at a rate faster than the ecosystems can sustain. The authors jointly evaluated the main environmental impacts associated with the emissions of these elements by using a set of indicators based on LCA, i.e., global warming, marine eutrophication, and freshwater eutrophication.

Hoehn et al. [124] used the energy return on investment (EROI) ratio, and a CE perspective, to develop an energy return on investment—circular economy index ($EROI_{ce}$) —to quantify the amount of nutritional energy recovered from the food loss in the Spanish food supply chain. The $EROI_{ce}$ index, based on a food waste-to-energy-to-food approach (the energy recovered from food loss is reintroduced into the food supply chain in form of food), was developed starting from the calculation of primary energy demand (PED) of each stage of the food supply chain under study. Here, we consider this index as a "life-cycle-based indicator" because it is based on the evaluating of energy flows along the entire food supply chain (agricultural production, processing, and packaging, distribution, and consumption).

Advancements in the assessment of CE strategies at the product level have been suggested by Niero and Kalbar [80], who coupled two sets of indicators via multi-criteria decision analysis, i.e., material circularity-based indicators—namely, material re-utilization score (MRS) and material circularity indicator (MCI)—and a selection of life cycle based-indicators (climate change, abiotic resource depletion, acidification, particulate matter, and water consumption). The MRS is the metric used to quantify material re-utilization developed by Cradle-to-Cradle Products Innovation Institute (C2C), while the MCI is the main index developed by the Ellen MacArthur Foundation (EMF) and Granta to measure how well a product performs in the CE context. The authors suggest exploring the application of the multicriteria analyses of life cycle-based indicators (including the socio-economic dimension) to address CE trade-offs and rebound effects. In a complementary manner, Schmidt Rivera et al. [81] proposed a set of indicators integrating techno-environmental and CE criteria to guide the design and development of new food packaging solutions within the new plastics economy. In detail, the authors considered nine indicators based on the CE guidelines developed by EMF, which focus on the materials and energy used in manufacturing and on end-of-life waste management, and four LCA based-indicators, i.e., climate change, depletion of fossil fuels, and metals, and primary energy demand to assess the environmental impacts of packaging from cradle to grave. Stanchev et al. [72] also developed an approach for measuring the material and environmental circularity performance of the anaerobic treatment of dairy processing effluents. Material CE performance was assessed by the "material circularity performance indicator" (MCPI), suggested by Agudelo-Vera et al. [125], which enables to evaluate to what extent the demand of resource

or energy flows reduced when the circularity loops are closed. On the other hand, the environmental performance was estimated by the "environmental circularity performance indicator" (ECPI) based on the ratio of the total environmental benefits and costs.

Combining LCA (global warming potential, acidification potential, eutrophication potential, and ReCIPE single score) and LCC (value-added) indicators, Laso et al. [46] suggested a method to assess the eco-efficiency of canned anchovy products with the eco-efficiency index (EEI), which enables the translation into economic terms of the environmental damage caused by the manufacture of a specific product.

Lokesh et al. [107] proposed a new set of hybridized sustainability indicators, drawn from the principles of green chemistry and resource (material and energy) circularity, to evaluate the environmental performance of bio-based products, bio-based packaging films, and mulch films in comparison with their commercial counterparts. These metrics are demonstrated via the application of a comparative LCA, incorporating the hybridized indicators including hazardous chemical use, circular-process feedstock intensity (CPFI), circular-process waste factor (CPWF), process material circularity (PMC), product renewability (PR), and circular-process energy intensity (CPEI). Through a set of LCA indicators, the authors also highlighted the resource and energy hotspots and toxicity to the environment and human health, and the quantification of impacts from the minimization of resources. Last but not least, Santagata et al. [126] used emergy-based circular economy indicators (not life cycle-based) to assess the sustainability of the urban eco-system. These indicators were developed by using Emergy accounting (EMA), which accounts for different categories of supporting contribution to the systems, including renewable and non-renewable energy and material resources, information and know-how, and finally labor and services. The authors consider EMA indicators as valuable tools to evaluate the implementation rate of CE patterns.

**Table 5.** Classification of reviewed circularity indicators according to Corona et al. (2020) [30].

| | Authors | Circularity Indices (Measuring the Circular Degree of a System) | CE Assessment Indicators (Assessing the Effects of Circularity) | | CE Application Level |
|---|---|---|---|---|---|
| | | | Life Cycle Based-Indicators | Other (No Life Cycle Based) | |
| 1 | Cobo et al. 2018a [45] | - Carbon circularity indicator (CIC)<br>- Nitrogen circularity indicator (CIN)<br>- Phosphorus circularity indicator (CIP) | - Global warming<br>- Marine eutrophication<br>- Freshwater eutrophication | - | Micro |
| 2 | Hoehn et al. 2019 [124] | - Energy return on investment-circular economy index (EROIce) | - Primary Energy Demand (PED) | - | Micro |
| 3 | Laso et al. 2018a [46] | - | - Global Warming Potential<br>- Acidification Potential<br>- Eutrophication Potential<br>- ReCIPE Single Score (SS)<br>- Value-added (VA) indicator<br>- Eco-efficiency index (EEI) | | Micro |

<div align="center"><b>Table 5.</b> <i>Cont.</i></div>

| Authors | Circularity Indices (Measuring the Circular Degree of a System) | CE Assessment Indicators (Assessing the Effects of Circularity) | | CE Application Level |
| | | Life Cycle Based-Indicators | Other (No Life Cycle Based) | |
|---|---|---|---|---|
| 4 Lokesh et al. 2020 [107] | - | - Global warming potential (GWP100), Respiratory inorganics, Human toxicity, Cancer, Acidification, Terrestrial and freshwater, Freshwater eutrophication, Water scarcity, Fossil resource depletion. <br> - Hazardous chemical use, Circular-process feedstock intensity (CPFI), Circular-process waste factor (CPWF), Process material circularity (PMC), Product renewability (PR), Circular-process energy intensity (CPEI). | - | Micro |
| 5 Niero and Kalbar, 2019 [80] | - Material Reutilization Score (MRS) <br> - Material Circularity Indicator (MCI) | - Climate Change, <br> - Abiotic Resource <br> - Depletion, <br> - Acidification, <br> - Particulate Matter <br> - Water Consumption | - | Micro |
| 6 Santagata et al. 2020 [126] | - | - | - Emergy Yield Ratio <br> - Environmental Loading Ratio <br> - Renewable fraction of emergy used <br> - Empower Density <br> - Population emergy intensity | Micro |
| 7 Schmidt Rivera et al. 2019 [81] | - Amount of material <br> - Mono or multi-components <br> - Recycling content <br> - Reuse rate <br> - Current waste management <br> - Current recycling rate <br> - Potential recyclability <br> - Use of renewable materials <br> - Use of renewable energy | - Climate change <br> - Depletion of fossil fuels <br> - Depletion of metals <br> - Primary energy demand (PED) | - | Micro |
| 8 Stanchev et al. 2020 [72] | - Material circularity performance indicator (MCPI) | - Environmental circularity performance indicator (ECPI) | - | Micro |

## 4. Discussion

Although it is not the only existing circularity indicator, it can undoubtedly be said that the "material circularity indicator" is one of the most robust tools for assessing the CE, and since the development of the original methodology published in 2015 by the EMF, the similarities between life cycle assessment and material circularity indicator have become evident. Both methodologies use system boundaries that encompass all phases of a product's life cycle, from creation to end of life. What differentiates the two approaches, however, is that the assessment of circularity cannot be limited to one life cycle because the circular pattern of one will inevitably influence the next. Slavishly quoting the methodology for measuring circularity, "the economic and environmental benefits from

using such materials do not commonly rest with the initial product but instead accrue through the successive use of the product or material over the multiple life cycles that they enable" [127] (p. 13). Therefore, the first step to marry these two methodologies should be to extend the boundaries of the system by integrating into the horizon of the analysis product losses, recycling and reuse in the next cycle, transport, and all processes that allow closing the loop of the LCA methodology according to a circular approach.

The literature review found that only 20% of studies use a cradle-to-grave [41,54] system boundary, only three of these explicitly refer to a cradle-to-cradle or system expansion approach [45,55,67] and only one of these integrate a circularity indicator [45]. Most studies focus on a partial system boundary and, therefore, life cycle analysis is more limited to assessing the environmental profile of the process or co-product [50].

Therefore, the studies do not aim at a true "circular strategy" because circularity is not really measured in most of them. Most articles use indicators relating to the use of material and energy resources but this is not enough to define the degree of circularity of a process or product, just as circularity alone cannot define sustainability.

LCA can assess the environmental impacts of a process and, through an eco-design approach, allows for the implementation of strategies to reduce these impacts, including a reduction in the use of resources and by considering the burden-shifting phenomenon whereby a change in one stage of the life cycle influences another one [128]. What attributional LCA cannot assess are rebound effects, a key element in sustainability assessments because it takes into account changes in production and consumption when the availability of a resource changes (positively or negatively) [129].

For these reasons, the assessment of circularity must pass through a multi-component approach that takes into account not only circularity itself but also other characteristic elements. The Ellen MacArthur Foundation identifies as complementary analyses, to the evaluation of circularity, the evaluation of risk factors such as material price variation, material supply chain, material scarcity and toxicity, and the evaluation of impact factors such as energy usage and $CO_2$ emissions, and water and toxicity [127].

The analysis of the papers shows the opposite situation, where the assessment of impact factors becomes the main driver for measuring sustainability [56], while circularity rather remains a goal to be achieved but hardly ever explicitly measured.

If we acknowledge the need to expand the boundaries of the system from a cradle-to-cradle perspective, at the same time, we should be aware that stating that a process serves to make the usual business model more circular does not automatically prove that this is the case—it has to be proven.

While the use of classic impact assessment methods, as already mentioned, can give us an assessment of resource and energy use, it does not allow a complete evaluation of circular strategies, which are often based on other fundamental factors such as product lifespan or functional unit, understood as the unit of measurement of product use. We can make the same product, using the same number of resources but, by increasing the efficiency of those resources by extending the life of the product, we can contribute positively to increasing the circularity of the process. This will probably not be detected by an LCA, especially if it does not take into account the use phase of the product.

This, moreover, may not apply to materials of biological origin, whose shelf life depends on factors not directly under anthropogenic control. However, in the case of food and agricultural products, extending the shelf life of products intended for consumption can make a significant contribution to the circularity of the process, reducing waste and thus increasing production efficiency. Moreover, as the results of the review show, lost food or products of biological origin that are no longer usable can easily be valorized in different ways, returning to the production cycle as fertilizers or generating bioenergy or bio-components with high added value.

The element that appears clear and seems to be unavoidable, however, is that an LCA complementary to a circularity assessment framework should always assess the whole life

cycle of a product and should consider its possible extensions, expanding also the time boundaries of the study by considering at least more than one life cycle.

A "circular LCA" [127] also take into account a detailed life cycle inventory analysis and consider "resource use" as one of mandatory impact indicator, or use specific supporting methodologies such as material flow analysis which takes into account the flows and stocks of materials and substances entering and leaving a defined system.

Some studies explore the use of methodologies in conjunction with LCA or standalone methodologies that take a life-cycle perspective. Above all, material flow analysis is a methodology that lends itself well as a supporting methodology for the assessment of circularity [36,118]. Other methodologies such as IO Analysis applied to the life cycle of a product allow the assessment of environmental impacts to be integrated with the impacts of positive or negative economic shocks and possible concatenated reactions on the economy [77].

As proposed by Santagata et al., [126] Emergy accounting method can also be a valuable support for measuring circularity. The authors of the paper also point out that conventional analysis methodologies (life cycle assessment, material flow accounting, cost–benefit analysis, etc.) do not capture all aspects of CE, proposing Emergy accounting as a possible solution; however, this methodology is also at the center of many scientific controversies and does not enjoy a consensus shared by the scientific community.

It might seem that the LCA methodology can make little contribution to the development of circularity metrics, but this is not the case because LCA is a well-established and standardized methodology while circularity assessment indicators are in development and their development is not in opposition to LCA but an ideal complement to it.

It should not be forgotten that talking about LCA is probably reductive because it should be part of a multi-objective framework (i.e., LCSA) aimed at analyzing the integrated sustainability of a process, product, system, or organization [29].

In this direction, as already discussed some studies have already explored the possibility of also using the life cycle costing methodology in the assessment of circularity [40]. While the main circularity indicators are essentially based on the increase in the utility of resources within an economic model, an approach that assesses the life cycle value flows of a product, process, system or organization is a fundamental complement to both circularity and sustainability assessment. What has already been discussed for the LCA also applies to the LCC, so the community of experts in life cycle methodologies and the CE must accompany the approach of these methodologies, resolving the technical and scientific issues [21] that have only briefly been discussed so far.

While the methodologies discussed so far mainly refer to environmental and/or economic metrics, the role that the large-scale economic model change will bring to the social level cannot be neglected. Therefore, the efforts of the scientific community must not neglect the social sustainability linked to the adoption of a new circular model. In addition, while the LCA and LCC methodologies are now scientifically established, the social counterpart of life cycle analysis, S-LCA is still in an embryonic state, so it is essential to manage the growth of this methodology so that it is consistent with new business models.

S-LCA is the latest tool developed within the family of life cycle methodologies. Since the 1990s, life cycle scholars felt the importance and urgency of taking into account social impacts [130], with an implied sustainability approach borrowed from the three pillars model, meaning that sustainability is composed of three dimensions (environment, economy, and society).

However, S-LCA is still not definitively codified in an agreed and consensus-based protocol. From its beginnings, a plethora of social impact assessment methods has been proposed for S-LCA, paying attention to the most diverse aspects, such as the social performances, the presence of hot spots, the accounting of risks, the consequences of a scenario, the externalities of a system, and the participation of [131,132]. Moreover, while LCA and LCC are focused on the impacts caused by the functioning of a system (regardless of the product or service it provides), very often, in S-LCA studies, the focus is

on companies' behaviors. This entails, therefore, evaluating a wide range of impacts, also not directly attributable to a life cycle (such as corruption, child labor, collective bargaining, fair wages, etc.). This is due to the epistemological eclecticism of social sciences, inevitably reflected in S-LCA studies [131]. Recently, the second edition of the S-LCA guidelines have been published [133], however, life cycle methodologies are still striving to reach an epistemological alignment, with LCA and LCC approaching impacts assessment in a post-positivist way, quantifying cause-effects relationships, and S-LCA mainly devoted to the interpretation of social performances according to stakeholders' perceptions and behaviors.

In general, many academics and scholars describe social sustainability as the most conceptually elusive pillar in sustainable development discourse [134].

Among the papers selected for this review, no one applies the S-LCA or another specific methodology for social impacts assessment. Rather, some degrees of social impact are explicitly associated, in some few cases, with economic performances, as is the case of [40,54,76]. The authors of [40,54] implemented the societal LCC to evaluate high-value products from food wastes (the former) and bio-based plastics (the latter). Societal LCC is a "welfare-economic", meaning that takes into account marketed goods and the effects on the society's welfare caused by exchanges that would otherwise not be accounted for, i.e., by identifying environmental externalities and measuring their relative monetary value [54]. In [76], the authors applied a hybrid life cycle assessment model to estimate social-economic impact through a multi-regional input-output database (Exiobase), with engineering process data of conventional and circular rice production systems. The indicators used in this case were "gross value added" and "employment" in each system, therefore, taking into account the "social significance" of economic performance. In their study, Monsiváis-Alonso et al. [44] defined social issues as "product or process-related aspects of operations that affect human safety and community welfare". The authors illustrated the possible social criteria suitable in the study of chemical/lipid processing, i.e., the satisfaction of social needs (SN), work satisfaction (WS), healthcare security coverage (HcS), employee turnover (EmpT), working hours (WH), employee complaints (EmpC), and risk assessment. However, none of these mentioned criteria is calculated in their study.

All the other papers reviewed only mention the importance of considering the social aspects of the life cycle, sometimes noting the social acceptance or desirability [50] and the social perception [67].

The lack of a specific, stand-alone, social impact assessment method in the reviewed papers here, but very often in sustainability assessments in general, allows us to reflect on the taxonomy and interconnections between "sustainability dimensions". Are they perfectly separable, or are they—at least—partly stackable? Do they have the same importance, or is there a hierarchy among them? These questions still remain open in the academic debate among life cycle scholars and practitioners.

## 5. Conclusions

This critical and systematic literature review provided a picture of the state-of-the-art of applications of the life cycle approach in the assessment of circularity of processes and products. Lights and shadows emerged—the relationship between circularity measurement and life cycle analyses still seem to be primordial, even though these two aspects are absolutely complimentary. CE measurement and life cycle methodologies should be similar to Manzoni's "The Betrothed"; as things stand, they seem almost as the protagonists of an arranged marriage who are getting to know each other in order to decide whether to continue their lives together or separate.

We now address the research questions of this study (Section 1.3), i.e., (1) how researchers apply LC methods to evaluate environmental, economic, and social consequences of agri-food circular processes?; (2) how LC methods are combined with other approaches in CE measuring?; and (3) have impact results been used to increase understanding of the sustainability implications of CE strategies?

In response to the first question, researchers do not fully exploit the possibilities offered by life cycle methodologies, limiting their application to a classical impact evaluation, disregarding the material quantification of circularity. Likewise, regarding the second question, very rarely the joint use of impact indicators and stand-alone circularity indicators is applied. It also emerged that the problem stays in the different views of the life cycle: in the case of impact evaluation, it is limited to cradle-to-gate or cradle-to-grave analyses; the circularity evaluation would require an extension of the system boundaries to more life cycles in a cradle-to-cradle perspective. Despite these limitations, it is evident how LC methodologies allow an improved understanding of the sustainability implications of CE strategies—they are not fully implemented or exploited to provide a circularity measure in a life cycle perspective; however, they allow evaluating environmental, economic, and social impacts of circular strategies (third research question).

The methodological development in this field is constantly evolving and new tools are increasingly being tested by the scientific community to identify the most effective ones and provide a measurement of circularity that takes into account sustainability issues. However, the scaffolding on which this methodological development must be based has already been built, and scholars cannot ignore it because there is a risk of making an irreparable mistake. In particular, experts in life cycle methodologies must strive to adopt some key elements to ensure that the results obtained fit perfectly with the measurements of circularity and that these can even be largely based on a common basis. The effort must also go in the direction of operability of the framework for measuring circularity and sustainability so that it does not have the opposite effect of an assessment structure that is so complex it is hardly usable, thus thwarting efforts to create new models of sustainable agri-food production and consumption.

**Supplementary Materials:** The following are available online at https://www.mdpi.com/2071-1050/13/5/2472/s1, Table S1. Completed review matrix (84 reviewed articles), Table S2. Fields of application, Table S3. Circularity topics.

**Author Contributions:** Conceptualization, T.S., A.I.D.L.; Formal analysis, T.S., N.I., G.F., A.I.D.L.; Methodology, T.S., N.I., G.F., A.I.D.L.; Data curation, T.S.; Project administration and supervision, A.I.D.L.; Validation, T.S., N.I., G.F.; Funding acquisition, A.I.D.L.; Writing—original draft, T.S., E.S., N.I., G.F., A.I.D.L.; Writing—review and editing, T.S., N.I., G.F., A.I.D.L. All authors have read and agreed to the published version of the manuscript.

**Funding:** This research was funded by PRIN 2017, DRASTIC (project code: 2017JYRZFF), Italian Ministry of Education, Universities, and Research (MIUR) within the research project "Driving the Italian Agri-Food System into a Circular Economy Model" (DRASTIC) PRIN 2017, (project code: 2017JYRZFF). This research is also co-funded by the European Commission, European Social Fund, and Calabria Region, through the post-doctoral research fellowship awarded to Iofrida. The authors are the only ones responsible for this scientific article; the European Commission and the Calabria Region decline any responsibility for the use that may be made of the information contained therein.

**Conflicts of Interest:** The authors declare no conflict of interest.

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
