# Peer review of "Sustainable Agri-Food Processes and Circular Economy Pathways in a Life Cycle Perspective: State of the Art of Applicative Research"

_sustainability, doi:10.3390/su13052472_

Round 1
Reviewer 1 Report
The manuscript is a systematic review of life cycle assessment based publications related to the subject of sustainable agri-food processes and circular pathways.
the manuscript appears to be well researched, with a concise reference list. Figure 2 summarises the approach in a transparent and easy to evaluate format. the comments below are minor.
Abstract line 19 - CE needs to be written in full
line 241 the sub heading 3.1 states 'Descriptive statistics' - i was unable to locate any statistical procedures in this section other than to describe the number of publications per year or by country. this is summary data but not a statistical analysis and therefore should be rephrased.
Figure 3 Publication trend by year - advise that the year 2021 is not included as this is not a complete year and makes the shape of the graph mis-leading
line 785 the authors note that there is a lack of a specific stand-alone social impact assessment method in the papers reviewed and assessments overall. There is then a bias towards the environmental side of the three pillars of sustainability (environment, economic, social)? Can they suggest a potential approach that could be used?
Reviewer 2 Report
Dear Authors,
The manuscript (sustainability-1113217) presented for review is very interesting and I recommend the article for publication in Sustainability Journal but after correction. In my opinion, the manuscript is a little bit too long, if the Authors try to shorter text will it be better for readers.
Reviewer
